# IMAGE DATASET FOR VISUAL OBJECTS CLASSIFICATION IN 3D PRINTING

**Hongjia Li,**\* **Xiaolong Ma,**\* **Zhe Li, Qiyuan An & Yanzhi Wang**
Department of Electrical Engineering and Computer Science
Syracuse University
Syracuse, NY 13244, USA
`{hli42,xma27,zli89,qan100,ywang393}@syr.edu`

**Aditya Singh Rathore,**\* **Chen Song & Wenyao Xu**
Department of Computer Science & Engineering
University at Buffalo, the State University of New York (SUNY)
Buffalo, NY 14260, USA
`{asrathor,csong5,wenyaoxu}@buffalo.edu`

## ABSTRACT

The rapid development in additive manufacturing (AM), also known as 3D printing, has brought about potential risk and security issues along with significant benefits. In order to enhance the security level of the 3D printing process, the present research aims to detect and recognize illegal components using deep learning. In this work, we collected a dataset of 61,340 2D images ($28 \times 28$ for each image) of 10 classes including guns and other non-gun objects, corresponding to the projection results of the original 3D models. To validate the dataset, we train a convolutional neural network (CNN) model for gun classification which can achieve 98.16% classification accuracy.

## 1 INTRODUCTION

The additive manufacturing (AM) technology, also known as 3D printing, has encountered significant advancement in the past decade and has become a major manufacturing process by many manufacturers. With high efficiency and convenience, 3D printing can be applied to various industry fields, such as medical, automotive, and aerospace engineering. However, the high personalization and low cost of such technique raise the chance that it may be used to fabricate illegal objects such as keys, guns, or even more powerful weapons. As a result, attention and concerns should be aroused by the security issues in 3D printing. This paper works towards this direction by integrating the detection for illegal weapons before or during the printing process, thereby enhancing the security level in 3D printing.

With the recent breakthrough of deep learning (LeCun et al., 2015), the visual object classification process using deep neural networks can be integrated in the 3D printing mechanism to recognize 3D objects from the projected grey-scale images at one or multiple viewpoints. Generally, a 3D printing chain can be divided into the cyber domain and the physical domain as shown in Figure 1. In the cyber domain, a object model is created through CAD (computer aided design) software and converted into the standard object file (STL), which describes only the surface geometry of the 3D object. Based on the STL file, the CAM (computer aided manufacturing) module slices the model into uniform layers and generates the toolpath file. The most widely used file format is G-code as described in wik (2018). The G-code contains all the information of the 3D digital design, includes the shape, the dimensions and volume. In the physical domain, the 3D printer conducts the physical manufacturing and fabricates the object.

To obtain the 2D images from sufficient viewpoints, we use the Ultimaker Cura software to display the 3D model and rotate in three dimensions. Based on the G-codes at various angles, we collect

---

\*H. Li, X. Ma and A. Rathore contributed equally to this work.

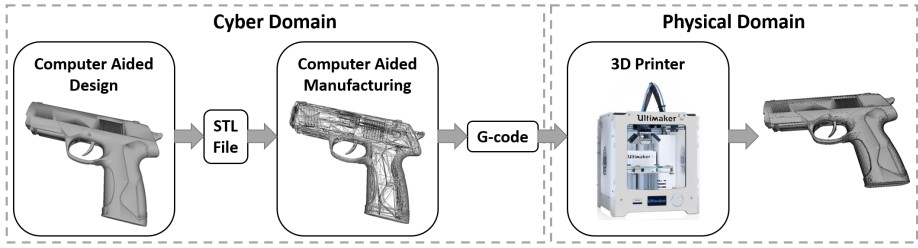

Figure 1: The overview of the 3D printing chain procedure of illegal objects.

61,340 2D greyscale images converted from the 3D model, and then re-size them into $28 \times 28$ to be compatible in size with the MNIST data set (LeCun et al., 1998). The transformation process includes the projections onto X-Y, Y-Z and X-Z planes. To maintain the best features in each image, we manually select the recognizable images in the dataset and discard the unrecognizable ones, which process is critical for the training process. Among the dataset, there are 51,840 training images and 9,500 test images. The dataset contains 10 classes, in which 4 classes are guns and the other 6 classes are randomly selected objects (including gun-like objects and toy guns). The data set size and number of categories are analogous to MNIST and CIFAR-10 data sets. In order to validate the dataset, we construct and train a deep convolutional neural network (CNN) with two convolutional layers for classification. According to the experimental results, the error rate can be reduced to 1.84% (i.e., we achieve classification accuracy of 98.16%). The data set and models are upon release to GitHub and the authors' websites.

## 2    THE DATASET CONSTRUCTION

This dataset is designed for use in 3D printing to address the security issues about the gun and illegal weapon control. In between the cyber domain and physical domain in the 3D printing chain, the integrated CNN can detect the gun at an early stage and terminate the manufacturing process. Additionally, this dataset can also be utilized for the research of using 2D images to recognize/reconstruct 3D objects, which can be widely applied in many fields such as drones/UAVs, self-driving and medical imaging systems.

The image dataset consists of 61,340 $28 \times 28$ greyscale images. It is comprised of 10 classes, 6,134 images for each class. The overall dataset is divided into 51,480 training images and 9,500 testing images. The 10 classes include pistol, revolver, special revolver, gun part, toy gun, gun-like object,

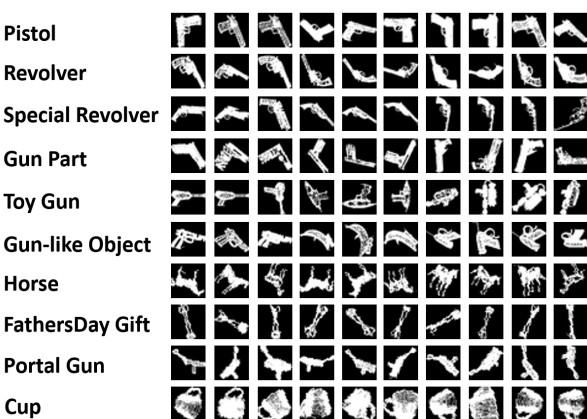

Figure 2: Illustration of the 10 classes in the dataset.

horse, fathersday gift, portal gun, and cup. The former 4 classes are classified as guns, and the rest are considered as not gun. The 10 classes are shown in Figure 2 with 10 sample images from each class.

The original 3D models contain 30 objects which are provided by MakerBot's Thingiverse (3dm). In these 30 models, different objects from various producers are included such as Beretta prop gun, Desert Eagle gun, and Smith & Wesson revolver. We pre-processed the models by clustering 30 models into 10 classes which can significantly improve the classification rate. For example, Beretta prop gun, Desert Eagle gun and PX4 are all considered as pistol; Ronald Ray gun, Space Dandy gun and Splator are all categorized into the toy gun class.

One major challenge on recognizing 3D objects using 2D images is the viewpoint variation. In order to overcome this limitation, during the transformation from STL files to G-code files, we use the Ultimaker Cura software (as described in (cur) online) to display the 3D models and rotate the models by 15 degree in each dimension, respectively. Using the obtained G-codes from different angles, 2D images are created through projection onto three planes, X-Y, Y-Z and X-Z. In order to maintain the most accurate features in the dataset, we manually select recognizable images from the three-plane projected results to be included in data set. Moreover, images with a large number of blank pixels exceeding a threshold are also excluded. In addition, the original images are post-processed with rotation, flipping and mirroring to increase the integrity and comprehensiveness of the dataset, which is similar to the building process of the CIFAR-10 data set.

## 3    DEEP NEURAL NETWORK MODEL

In image classification, multiple deep neural network models have been implemented with excellent performance, such as convolutional nets developed by LeCun et al. (1998), AlexNet constructed by Krizhevsky et al. (2012) , and GoogLeNet, a 22 layers deep network proposed by Szegedy et al. (2015).

To address the object classification problem for 3D printing, we construct and train a classic CNN model with two convolutional layers to validate the dataset. The CNN model is shown in Figure 3. In our approach, we use the standard GNU zip compression algorithm to compress the dataset into .gz file as the input for the neural network. We can achieve up to 98.16% accuracy in the classification problem (with 1.84% error rate).

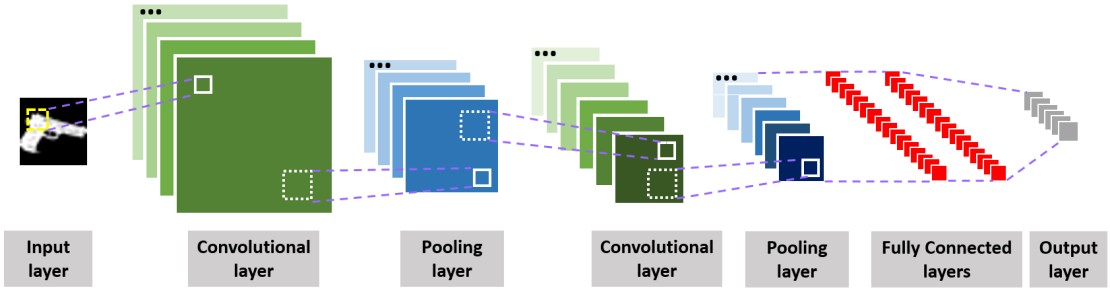

Figure 3: The convolutional neural network model to solve the object classification problem in 3D printing.

## 4    CONCLUSION

In this work, we collected a image dataset for gun classification and security enhancing in the 3D printing process. In this 10-class dataset, there are 61,340 28×28 greyscale images converted from 3D models at various viewpoints. The dataset is divided into 51,480 training images and 9,500 test images. A deep convolutional neural network is constructed to validate the dataset and the error rate for classifying guns can be reduced to 1.84 percent.

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
