# OpenReview forum: "Image Dataset for Visual Objects Classification in 3D Printing"
_ICLR.cc/2018/Workshop — Reject_

### Official Review · AnonReviewer3 · 2018-03-04
**An important problem, but no ML contribution**

**Rating:** 4
**Confidence:** 5

**Review:**

The paper deals with an important problem of cyber-security of 3D printing. Despite the solid motivation, the data set does not really have any significance for the ML community (given that the data set does not seem to pose significant difficulty and many similar 2D data sets corresponding to 3D models for training models such as multi-view CNN are already available). It will probably be still very relevant for the manufacturing community. However, it was not clear why 2D data was collected instead of the 3D models. Recent work has shown that 3D CNNs handle 3D CAD models for various manufacturing problems. A 3D database would actually be more useful that way for even the ML community.

---

### Official Review · AnonReviewer1 · 2018-03-09
**New 2D grey scale image Dataset**

**Rating:** 5
**Confidence:** 5

**Review:**

This paper presents a new dataset of 2D grey scale images. The dataset is a collection of 61,340 pictures (28x28 pixels) of 10 object categories. The categories include different types of gun and gun parts and other gun-alike objects. The motivaton of the dataset is to enhance the security of 3D printing process by detecting automatically the impression of illegal objects, like guns.

Alhough the motivation of the dataset is interesting, the dataset itself is not chanllenging and does not open any new research problem. The authors test a standard CNN arquitecture for object classification and obtain a classification accuracy of 98.16%.

In my opinion this work is interesting but it can not be accepted as a research paper. The dataset can not be considered as a scientific contribution and there is no technical contribution.

---

### Official Review · AnonReviewer2 · 2018-03-10
**no new research**

**Rating:** 4
**Confidence:** 4

**Review:**

The paper introduces a dataset having 3d models of guns and similar objects rendered from many different viewpoints. The motivation is to have 3D printers that refuse to print certain dangerous objects. This is an interesting application but in terms of research there isn't much in the paper. Rendering a few cad models and training a convnet to distinguish them from many viewpoints has been the subject of many papers -- for example see references in http://modelnet.cs.princeton.edu/

---

### Decision · Program_Chairs · 2018-03-20
**ICLR 2018 Workshop Acceptance Decision**

**Decision:**

Reject

**Comment:**

Based on the reviews, this paper has not been accepted for presentation at the ICLR workshop. However, the conversation and updates can continue to appear here on OpenReview.